# CLEAR: Generative Counterfactual Explanations on Graphs

**Jing Ma**
University of Virginia
Charlottesville, VA, USA
jm3mr@virginia.edu

**Ruocheng Guo**
Bytedance AI Lab
London, UK
rguo.asu@gmail.com

**Saumitra Mishra**
J.P. Morgan AI Research
London, UK
saumitra.mishra@jpmorgan.com

**Aidong Zhang**
University of Virginia
Charlottesville, VA, USA
aidong@virginia.edu

**Jundong Li**
University of Virginia
Charlottesville, VA, USA
jundong@virginia.edu

## Abstract

Counterfactual explanations promote explainability in machine learning models by answering the question "*how should an input instance be perturbed to obtain a desired predicted label?*". The comparison of this instance before and after perturbation can enhance human interpretation. Most existing studies on counterfactual explanations are limited in tabular data or image data. In this work, we study the problem of counterfactual explanation generation on graphs. A few studies have explored counterfactual explanations on graphs, but many challenges of this problem are still not well-addressed: 1) optimizing in the discrete and disorganized space of graphs; 2) generalizing on unseen graphs; and 3) maintaining the causality in the generated counterfactuals without prior knowledge of the causal model. To tackle these challenges, we propose a novel framework CLEAR which aims to generate counterfactual explanations on graphs for graph-level prediction models. Specifically, CLEAR leverages a graph variational autoencoder based mechanism to facilitate its optimization and generalization, and promotes causality by leveraging an auxiliary variable to better identify the underlying causal model. Extensive experiments on both synthetic and real-world graphs validate the superiority of CLEAR over the state-of-the-art methods in different aspects.

## 1 Introduction

To facilitate explainability in opaque machine learning (ML) models (e.g., how predictions are made by a model, and what to do to achieve a desired outcome), explainable artificial intelligence (XAI) [1] has recently attracted significant attention in many communities. Among the existing work of XAI, a special class, i.e., *counterfactual explanation* (CFE) [2], promotes model explainability by answering the following question: "*For a specific instance, how should the input features $X$ be slightly perturbed to new features $X'$ to obtain a different predicted label (often a desired label) from ML models?*". The original instance whose prediction needs to be explained is called an *explainee instance*, and the generated instances after perturbation are referred to as "counterfactual explanations". Generally, CFE promotes human interpretation through the comparison between $X$ and $X'$. With its intuitive nature, CFEs can be deployed in various real-world scenarios such as loan application and legal framework [3]. Different from traditional CFE studies [2, 3, 4, 5] on tabular or image data, recently, CFE on graphs is also an emerging field in many domains with graph structure data such as molecular analysis [6] and professional networking [7]. For example, consider a grant application prediction [8] model with each input instance as a graph representing a research team's collaboration network,

36th Conference on Neural Information Processing Systems (NeurIPS 2022).

where each node represents a team member, and each edge signifies a collaboration relationship between them. Team leaders can improve their teams for next application by changing the original graph according to the counterfactual with a desired predicted label (application being granted). If the counterfactual is more dense than the original, the team leader may then encourage more team collaborations. To this end, in this work, we investigate the problem of generating counterfactual explanations on graphs. As shown in Fig. 1, given a prediction model $f$ on graphs, for a graph instance $G$, we aim to generate counterfactuals (e.g., $G^{CF}$) which are slightly different from $G$ w.r.t. their node features or graph structures to elicit a desired model prediction. Specifically, we focus on graph-level prediction without any assumptions of the prediction model type and its model access, i.e., $f$ can be a black box with unknown structure.

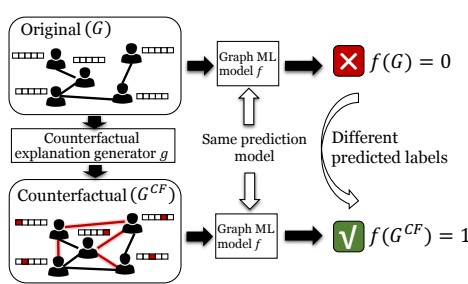

Figure 1: An example of CFE on graphs.

Recently, a few studies [9, 6, 10, 11, 12, 13] explore to extend CFEs into graphs. However, this problem still remains a daunting task due to the following key challenges: 1) **Optimization:** Different from traditional data, the space of perturbation operations on graphs (e.g., add/remove nodes/edges) is discrete, disorganized, and vast, which brings difficulties for optimization in CFE generation. Most existing methods [6, 11] search for graph counterfactuals by enumerating all the possible perturbation operations on the current graph. However, such enumeration on graphs is of high complexity, and it is also challenging to solve an optimization problem in such a complex search space. Few graph CFE methods which enable gradient-based optimization either rely on domain knowledge [6] or assumptions [10] about the prediction model to facilitate optimization. However, these knowledge and assumptions limit their applications in different scenarios. 2) **Generalization:** The discrete and disorganized nature of graphs also brings challenges for the generalization of CFE methods on unseen graphs, as it is hard to sequentialize the process of graph CFE generation and then generalize it. Most existing CFE methods on graphs [9, 12] solve an optimization problem for each explainee graph separately. These methods, however, cannot be generalized to new graphs. 3) **Causality:** It is challenging to generate counterfactuals that are consistent with the underlying causality. Specifically, causal relations may exist among different node features and the graph structure. In the aforementioned example, for each team, after establishing more collaborations, the team culture may be causally influenced. Incorporating causality can generate more realistic and feasible counterfactuals [4], but most existing CFE methods either cannot handle causality, or require too much prior knowledge about the causal relations in data.

To address the aforementioned challenges, in this work, we propose a novel framework — generative **C**ounterfactua**L** **E**xpl**A**nation gene**R**ator for graphs (CLEAR). At a high level, CLEAR is a generative, model-agnostic CFE generation framework for graph prediction models. For any explainee graph instance, CLEAR aims to generate counterfactuals with slight perturbations on the explainee graph to elicit a desired predicted label, and the counterfactuals are encouraged to be in line with the underlying causality. More specifically, to facilitate the optimization of the CFE generator, we map each graph into a latent representation space, and output the counterfactuals as a probabilistic fully-connected graph with node features and graph structure similar as the explainee graph. In this way, the framework is differentiable and enables gradient-based optimization. To promote generalization of the CFE generator on unseen graphs, we propose a generative way to construct the counterfactuals. After training the CFE generator, it can be efficiently deployed to generate (multiple) counterfactuals on unseen graphs, rather than retraining from scratch. To generate more realistic counterfactuals without explicit prior knowledge of the causal relations, inspired by the recent progress in nonlinear independent component analysis (ICA) [14] and its connection with causality [15], we make an exploration to promote causality in counterfactuals by leveraging an auxiliary variable to better identify the latent causal relations. The main contributions of this work can be summarized as follows: 1) **Problem.** We study an important problem: counterfactual explanation generation on graphs. We analyze its challenges including optimization, generalization, and causality. To the best of our knowledge, this is the first work jointly addressing all these challenges of this problem. 2) **Method.** We propose a novel framework CLEAR to generate counterfactual explanations for graphs. CLEAR can generalize to unseen graphs, and promote causality in the counterfactuals. 3) **Experiments.** We conduct extensive experiments on both synthetic and real-world graphs to validate the superiority of our method over state-of-the-art baselines of graph CFE generation.

## 2 Preliminaries

A graph $G = (X, A)$ is specified with its node feature $X$ and adjacency matrix $A$. We have a graph prediction model $f : \mathcal{G} \to \mathcal{Y}$, where $\mathcal{G}$ and $\mathcal{Y}$ represent the space of graphs and labels, respectively. In this work, we assume that we can access the prediction of $f$ for any input graph, but we do not assume the access of any knowledge of the prediction model itself. For a graph $G \in \mathcal{G}$, we denote the output of the prediction model as $Y = f(G)$. A counterfactual $G^{CF} = (X^{CF}, A^{CF})$ is expected to be similar as the original explainee graph $G$, but the predicted label for $G^{CF}$ made by $f$ (i.e., $Y^{CF} = f(G^{CF})$) should be different from $f(G)$. With a desired label $Y^*$ (here $Y^* \neq Y$), the counterfactual $G^{CF}$ is considered to be *valid* if and only if $Y^* = Y^{CF}$. In this paper, we mainly focus on graph classification, but our framework can also be extended to other tasks such as node classification, as discussed in Appendix D.

Suppose we have a set of graphs sampled from the space $\mathcal{G}$, and different graphs may have different numbers of nodes and edges. A counterfactual explanation generator can generate counterfactuals for any input graph $G$ w.r.t. its desired predicted label $Y^*$. As aforementioned, most existing CFE methods on graphs have limitations in three aspects: optimization, generalization, and causality. Next, we provide more background of causality. The causal relations between different variables (e.g., node features, degree, etc.) in the data can be described with a structural causal model (SCM):

**Definition 1.** *(Structural Causal Model) A structural causal model (SCM) [15] is denoted by a triple $(U, V, F)$: $U$ is a set of exogenous variables, and $V$ is a set of endogenous variables. The structural equations $F = \{F_1, ..., F_{|V|}\}$ determine the value for each $V_i \in V$ with $V_i = F_i(\texttt{PA}_i, U_i)$, here $\texttt{PA}_i \subseteq V \backslash V_i$ denotes the "parents" of $V_i$, and $U_i \subseteq U$.*

**Definition 2.** *(Causality in CFE) For an explainee graph $G$, a counterfactual $G^{CF}$ satisfies causality if the change from $G$ to $G^{CF}$ is consistent with the underlying structural causal model.*

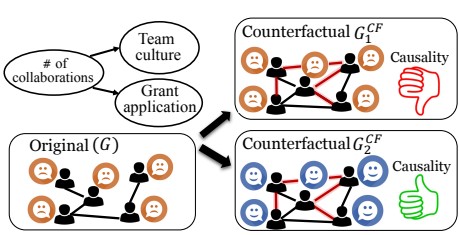

Figure 2: An example of causality in CFE.

**Example:** In the aforementioned grant application example, for a research team which has been rejected for an application before (as the graph $G$ in Fig. 2), to get the next application approved, a valid counterfactual may suggest this team to improve the number of collaborations between team members. Based on real-world observations, we assume that an additional causal relation exists: the number of team collaborations causally affects the team culture. For example, for the same team, if more collaborations had been established, then the team culture should have been improved in terms of better member engagement and respect for diversity. The SCM is illustrated in Fig. 2, where we leave out the exogenous variables for simplicity. Although the team culture usually does not affect the result of grant application, the counterfactuals with team culture changed correspondingly when the number of collaborations changes are more consistent with the ground-truth SCM. $G_2^{CF}$ in Fig. 2 shows an example of such counterfactuals. In contrast, if a counterfactual improves a team's number of collaborations alone without improving the team culture (see $G_1^{CF}$ in Fig. 2), then it violates the causality. As discussed in [4], traditional CFE methods often optimize on a single instance, and are prone to perturb different features independently, thus they often fail to satisfy the causality.

## 3 The Proposed Framework — CLEAR

In this section, we describe a novel generative framework — CLEAR, which addresses the problem of counterfactual explanation generation for graphs. First, we introduce its backbone CLEAR-VAE to enable optimization on graphs and generalization on unseen graph instances. This backbone is based on a graph variational auto-encoder (VAE) [16] mechanism. On top of CLEAR-VAE, we then promote the causality of CFEs with an auxiliary variable.

### 3.1 CLEAR-VAE: Backbone of Graph Generative Counterfactual Explanations

Different from most existing methods [9, 12] for CFE generation on graphs which focus on a single graph instance, CLEAR is based on a generative backbone CLEAR-VAE which can efficiently

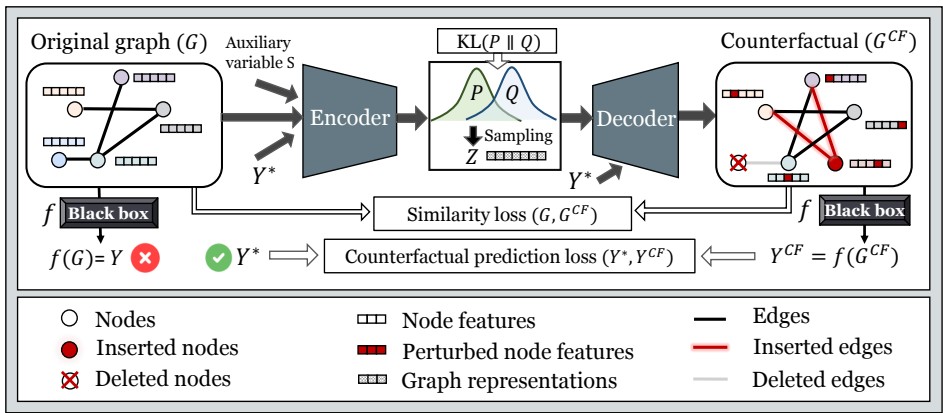

Figure 3: An illustration of the proposed framework CLEAR.

generate CFEs for different graphs after training, even for the graphs that do not appear in the training data. As shown in Fig. 3, CLEAR-VAE follows a traditional graph VAE [16] architecture with an encoder and a decoder. The encoder maps each original graph $G = (X, A)$ into a latent space as a representation $Z$, then the decoder generates a counterfactual $G^{CF}$ based on the latent representation $Z$. Following the VAE mechanism as in [16, 17] and its recent application in CFE generation for tabular data [4, 18], the optimization objective is based on the evidence lower bound (ELBO), which is a lower bound for the log likelihood $\ln P(G^{CF}|Y^*, G)$. Here, $P(G^{CF}|Y^*, G)$ is the probability of the generated counterfactual $G^{CF}$ conditioned on an input explainee graph $G$ and a desired label $Y^*$. The ELBO for CLEAR-VAE can be derived as follows:

$$\ln P(G^{CF}|Y^*, G) \geq \mathbb{E}_Q[\ln P(G^{CF}|Z, Y^*, G)] - \text{KL}(Q(Z|G, Y^*)\|P(Z|G, Y^*)), \quad (1)$$

where $Q$ refers to the approximate posterior distribution $Q(Z|G, Y^*)$, and $\text{KL}(\cdot\|\cdot)$ means the Kullback-Leibler (KL) divergence. The first term $P(G^{CF}|Z, Y^*, G)$ denotes the probability of the generated counterfactual conditioned on the representation $Z$ and the desired label $Y^*$. Due to the lack of ground-truth counterfactuals, it is hard to directly optimize this term. But inspired by [4], maximizing this term can be considered as generating valid graph counterfactuals w.r.t. the desired label $Y^*$, thus we replace this term with $-\mathbb{E}_Q[d(G, G^{CF}) + \alpha \cdot l(f(G^{CF}), Y^*)]$, where $d(\cdot, \cdot)$ is a similarity loss, which is a distance metric to measure the difference between $G$ and $G^{CF}$, $l(\cdot)$ is the counterfactual prediction loss to measure the difference between the predicted label $f(G^{CF})$ and the desired label $Y^*$. In summary, these two terms encourage the model to output counterfactuals which are similar as the input graph but elicit the desired predicted label. $\alpha$ is a hyperparameter to control the weight of the counterfactual prediction loss. Overall, the loss function of CLEAR-VAE is:

$$\mathcal{L} = \mathbb{E}_Q[d(G, G^{CF}) + \alpha \cdot l(f(G^{CF}), Y^*)] + \text{KL}(Q(Z|G, Y^*)\|P(Z|G, Y^*)). \quad (2)$$

**Encoder.** In the encoder, the input includes node features $X$ and graph structure $A$ of the explainee graph $G$, as well as the desired label $Y^*$, the output is latent representation $Z$. The encoder learns the distribution $Q(Z|G, Y^*)$. We use a Gaussian distribution $P(Z|G, Y^*) = \mathcal{N}(\mu_z(Y^*), \text{diag}(\sigma_z^2(Y^*)))$ as prior, and enforce the learned distribution $Q(Z|G, Y^*)$ to be close to the prior by minimizing their KL divergence. Here, $\mu_z(Y^*)$ and $\text{diag}(\sigma_z^2(Y^*))$ are mean and diagonal covariance of the prior distribution learned by a neural network module. $Z$ is sampled from the learned distribution $Q(Z|G, Y^*)$ with the widely-used reparameterization trick [17].

**Decoder.** In the decoder, the input includes $Z$ and $Y^*$, while the output is the counterfactual $G^{CF} = (X^{CF}, A^{CF})$. Different counterfactuals can be generated for one explainee graph by sampling $Z$ from $Q(Z|G, Y^*)$ for multiple times. The adjacency matrix is often discrete, and typically assumed to include only binary values ($A_{(i,j)} = 1$ if edge from node $i$ to node $j$ exists, otherwise $A_{(i,j)} = 0$). To facilitate optimization, inspired by recent graph generative models [9, 16], our decoder outputs a probabilistic adjacency matrix $\hat{A}^{CF}$ with elements in range $[0, 1]$, and then generates a binary adjacency matrix $A^{CF}$ by sampling from Bernoulli distribution with probabilities in $\hat{A}^{CF}$. We calculate the similarity loss in Eq. (2) as:

$$d(G, G^{CF}) = d_A(A, \hat{A}^{CF}) + \beta \cdot d_X(X, X^{CF}), \quad (3)$$

where $d_A$ and $d_X$ are metrics to measure the distance between two graphs w.r.t. their graph structures and node features, respectively. $\beta$ controls the weight for the similarity loss w.r.t. node features. More details of model implementation are in Appendix B.

## 3.2 CLEAR: Improving the Causality in Counterfactual Explanations

To further incorporate the causality in the generated CFEs, most existing studies [4, 19, 20] leverage certain prior knowledge (e.g., a given path diagram which depicts the causal relations among variables) of the SCM. However, it is often difficult to obtain sufficient prior knowledge of the SCM in real-world data, especially for graph data. In this work, we do not assume the access of the prior knowledge of SCM, and only assume that the observational data is available. However, the key challenge, as shown in [19], is that it is impossible to *identify* the ground-truth SCM from observational data without additional assumptions w.r.t. the structural equations and the exogenous variables, because different SCMs may result in the same observed data distribution. Considering that different SCMs can generate different counterfactuals, the identifiability of SCM is an obstacle of promoting causality in CFE. Fortunately, enlightened by recent progress in nonlinear independent component analysis (ICA) [14, 21], we make an initial exploration to promote causality in CFE by improving the identifiability of the latent variables in our CFE generator with the help of an auxiliary observed variable. This CFE generator is denoted by CLEAR.

In nonlinear independent component analysis (ICA) [14, 21, 22, 23], it is assumed that the observed data, e.g., $X$, is generated from a smooth and invertible nonlinear transformation of independent latent variables (referred to as *sources*) $Z$. Identifying the sources and the transformation are the key goals in nonlinear ICA. Similarly, traditional VAE models also assume that the observed features $X$ are generated by a set of latent variables $Z$. However, traditional VAEs cannot be directly used for nonlinear ICA as they lack identifiability, i.e., we can find different $Z$ that lead to the same observed data distribution $p(X)$. Recent studies [14] have shown that the identifiability of VAE models can be improved with an auxiliary observed variable $S$ (e.g., a time index or class label), which enables us to use VAE for nonlinear ICA problem. As discussed in [22, 23], a SCM can be considered as a nonlinear ICA model if the exogenous variables in the SCM are considered as the sources in nonlinear ICA. Similar connections can be built between the structural equations in SCM and the transformations in ICA. Such connections shed a light on improving the identifiability of the underlying SCM without much explicit prior knowledge of the SCM.

With this idea, based on the backbone CLEAR-VAE, CLEAR improves the causality in counterfactuals by promoting identifiability with an observed auxiliary variable $S$. Intuitively, we expect the graph VAE can capture the exogenous variables of the SCM in its representations $Z$, and approximate the data generation process from the exogenous variables to the observed data, which is consistent with the SCM. Here, for each graph, the auxiliary variable $S$ can provide additional information for CLEAR to better identify the exogenous variables in the SCM, and thus can elicit counterfactuals with better causality. To achieve this goal, following the previous work of nonlinear ICA [14, 21], we make the following assumption:

**Assumption 1.** *We assume that the prior on the latent variables $P(Z|S)$ is conditionally factorial.*

With this assumption, the original data can be stratified by different values of $S$, and each separated data stratum can be considered to be generated by the ground-truth SCM under certain constraints (e.g., the range of values that the exogenous variables can take). When the constraints become more restricted, the space of possible SCMs that can generate the same data distribution in each data stratum shrinks. In this way, the identification of the ground-truth SCM can be easier if we leverage the auxiliary variable $S$. Here, with the auxiliary variable $S$, we infer the ELBO of CLEAR:

**Theorem 1.** The evidence lower bound (ELBO) to optimize the framework CLEAR is:

$$\ln P(G^{CF}|S, Y^*, G) \geq \mathbb{E}_Q[\ln P(G^{CF}|Z, S, Y^*, G)] - \mathrm{KL}(Q(Z|G, S, Y^*) \| P(Z|G, S, Y^*)), \quad (4)$$

the detailed proof is shown in Appendix A.

**Loss function of CLEAR.** Based on the above ELBO, the final loss function of CLEAR is:

$$\mathcal{L} = \mathbb{E}_Q[d(G, G^{CF}) + \alpha \cdot l(f(G^{CF}), Y^*)] + \mathrm{KL}(Q(Z|G, S, Y^*) \| P(Z|G, S, Y^*)). \quad (5)$$

**Encoder and Decoder.** The encoder takes the input $G$, $S$, and $Y^*$, and outputs $Z$ as the latent representation. We use a Gaussian prior $P(Z|G, S, Y^*) = \mathcal{N}(\mu_z(S, Y^*), \mathrm{diag}(\sigma_Z^2(S, Y^*))$ with its

mean and diagonal covariance learned by neural network, and we encourage the learned approximate posterior $Q(Z|G, S, Y^*)$ to approach the prior by minimizing their KL divergence. Similar to the backbone CLEAR-VAE, the decoder takes the inputs $Z$ and $Y^*$ to generate one or multiple counterfactuals $G^{CF}$ for each explainee graph. More implementation details are in Appendix B.

# 4 Experiment

In this section, we evaluate our framework CLEAR with extensive experiments on both synthetic and real-world graphs. In particular, we answer the following research questions in our experiments: **RQ1:** How does CLEAR perform compared to state-of-the-art baselines? **RQ2:** How do different components in CLEAR contribute to the performance? **RQ3:** How can the generated CFEs promote model explainability? **RQ4:** How does CLEAR perform under different settings of hyperparameters?

## 4.1 Baselines

We use the following baselines for comparison: 1) **Random**: For each explainee graph, it randomly perturbs the graph structure for at most $T$ steps. Stop if a desired predicted label is achieved. 2) **EG-IST**: For each explainee graph, it randomly inserts edges into it for at most $T$ steps. 3) **EG-RM**: For each explainee graph, it randomly removes edges for at most $T$ steps. 4) **GNNExplainer**: GNNExplainer [12] is proposed to identify the most important subgraphs for prediction. We apply it for CFE generation by removing the important subgraphs identified by GNNExplainer. 5) **CF-GNNExplainer**: CF-GNNExplainer [9] is proposed for generating counterfactual ego networks in node classification tasks. We adapt CF-GNNExplainer for graph classification by taking the whole graph (instead of the ego network of any specific node) as input, and optimizing the model until the graph classification (instead of node classification) label has been changed to the desired one. 6) **MEG**: MEG [6] is a reinforcement learning based CFE generation method. In all the experiments, we set $T = 150$. More details of the baseline setup can be referred in Appendix B.

## 4.2 Datasets

We evaluate our method on three datasets, including a synthetic dataset and two datasets with real-world graphs. **(1) Community.** This dataset contains synthetic graphs generated by the Erdös-Rényi (E-R) model [24]. In this dataset, each graph consists of two 10-node communities. The label $Y$ is determined by the average node degree in the first community (denoted by $\deg_1(A)$). According to the causal model (in Appendix B), when $\deg_1(A)$ increases (decreases), the average node degree in the second community $\deg_2(A)$ should decrease (increase) correspondingly. We take this causal relation $\deg_1(A) \to \deg_2(A)$ as our causal relation of interest, and denote it as $R$. Correspondingly, we define a causal constraint for later evaluation of causality: "$(\deg_1(A^{CF}) > \deg_1(A)) \Rightarrow (\deg_2(A^{CF}) < \deg_2(A))$" OR "$(\deg_1(A^{CF}) < \deg_1(A)) \Rightarrow (\deg_2(A^{CF}) > \deg_2(A))$". **(2) Ogbg-molhiv.** In this dataset, each graph stands for a molecule, where each node represents an atom, and each edge is a chemical bond. As the ground-truth causal model is unavailable, we simulate the label $Y$ and causal relation of interest $R$. **(3) IMDB-M.** This dataset contains movie collaboration networks from IMDB. In each graph, each node represents an actor or an actress, and each edge represents the collaboration between two actors or actresses in the same movie. Similarly as the above datasets, we simulate the label $Y$ and causal relation of interest $R$, and define causal constraints corresponding to $R$. It is worth mentioning that the causal relation of interest $R$ in the three datasets covers different types of causal relations respectively: i) causal relations between variables in graph structure $A$; ii) between variables in node features $X$; iii) between variables in $A$ and in $X$. Thereby we comprehensively evaluate the performance of CLEAR in leveraging different modalities (node features and graph structure) of graphs to fit in different types of causal relations. More details about datasets are in Appendix B.

## 4.3 Evaluation Metrics

**Validity:** the proportion of counterfactuals which obtain the desired labels.

$$\text{Validity} = \frac{1}{N} \sum_{i \in [N]} \frac{1}{N^{CF}} \sum_{j \in [N^{CF}]} |\mathbf{1}(f(\mathbf{G}_{(i,j)}^{CF}) = y_i^*)|, \tag{6}$$

where $N$ is the number of graph instances, $N^{CF}$ is the number of counterfactuals generated for each graph. $\mathbf{G}_{(i,j)}^{CF} = (\mathbf{X}_{(i,j)}^{CF}, \mathbf{A}_{(i,j)}^{CF})$ denotes the $j$-th counterfactual generated for the $i$-th graph instance. Here, $y_i^*$ is the realization of $Y^*$ for the $i$-th graph. $\mathbf{1}(\cdot)$ is an indicator function which outputs 1 when the input condition is true, otherwise it outputs 0.

Table 1: The performance (mean ± standard deviation over ten repeated executions) of different methods of CFEs on graphs. The best results are in bold, and the runner-up results are underlined.

| Datasets | Methods | Validity (↑) | Proximity$_X$ (↑) | Proximity$_A$ (↑) | Causality (↑) | Time (↓) |
|---|---|---|---|---|---|---|
| Community | Random | 0.53 ± 0.05 | N/A | 0.77 ± 0.02 | 0.52 ± 0.06 | 0.20 ± 0.01 |
| | EG-IST | 0.53 ± 0.05 | N/A | 0.66 ± 0.03 | 0.13 ± 0.06 | 0.27 ± 0.03 |
| | EG-RMV | 0.55 ± 0.04 | N/A | **0.85 ± 0.01** | 0.03 ± 0.02 | 0.15 ± 0.01 |
| | GNNExplainer | 0.52 ± 0.06 | N/A | 0.71 ± 0.01 | 0.05 ± 0.00 | 2.87 ± 0.08 |
| | CF-GNNExplainer | 0.90 ± 0.04 | N/A | 0.72 ± 0.00 | 0.14 ± 0.02 | 25.14 ± 1.22 |
| | MEG | 0.88 ± 0.04 | N/A | 0.71 ± 0.01 | 0.10 ± 0.03 | 27.29 ± 1.32 |
| | **CLEAR(ours)** | **0.94 ± 0.02** | **0.91 ± 0.01** | 0.77 ± 0.00 | **0.65 ± 0.03** | **0.01 ± 0.01** |
| Ogbg-molhiv | Random | 0.48 ± 0.09 | N/A | 0.87 ± 0.02 | 0.46 ± 0.1 | 0.17 ± 0.02 |
| | EG-IST | 0.48 ± 0.09 | N/A | 0.83 ± 0.03 | 0.46 ± 0.09 | 0.19 ± 0.04 |
| | EG-RM | 0.483 ± 0.09 | N/A | **0.96 ± 0.01** | 0.47 ± 0.09 | 0.17 ± 0.04 |
| | GNNExplainer | 0.50 ± 0.01 | N/A | 0.92 ± 0.00 | 0.48 ± 0.10 | 2.78 ± 0.10 |
| | CF-GNNExplainer | 0.54 ± 0.02 | N/A | 0.92 ± 0.01 | 0.49 ± 0.02 | 27.93 ± 1.20 |
| | MEG | 0.49 ± 0.03 | N/A | 0.93 ± 0.01 | 0.50 ± 0.10 | 22.39 ± 2.20 |
| | **CLEAR(ours)** | **0.98 ± 0.01** | **0.92 ± 0.02** | 0.95 ± 0.01 | **0.64 ± 0.02** | **0.01 ± 0.00** |
| IMDB-M | Random | 0.50 ± 0.04 | N/A | 0.67 ± 0.01 | 0.43 ± 0.08 | 0.19 ± 0.01 |
| | EG-IST | 0.56 ± 0.12 | N/A | 0.67 ± 0.06 | 0.45 ± 0.07 | 0.16 ± 0.03 |
| | EG-RM | 0.45 ± 0.11 | N/A | **0.75 ± 0.03** | 0.53 ± 0.08 | 0.18 ± 0.02 |
| | GNNExplainer | 0.43 ± 0.10 | N/A | 0.62 ± 0.02 | 0.50 ± 0.02 | 2.46 ± 0.50 |
| | CF-GNNExplainer | 0.95 ± 0.02 | N/A | 0.74 ± 0.02 | 0.51 ± 0.02 | 22.21 ± 1.42 |
| | MEG | 0.90 ± 0.02 | N/A | 0.72 ± 0.02 | 0.51 ± 0.02 | 24.12 ± 1.08 |
| | **CLEAR(ours)** | **0.96 ± 0.01** | **0.99 ± 0.00** | 0.75 ± 0.01 | **0.73 ± 0.01** | **0.01 ± 0.00** |

**Proximity:** the similarity between the generated counterfactuals and the input graph. Specifically, we separately evaluate the proximity w.r.t. node features and graph structure, respectively.

$$\text{Proximity}_X = \frac{1}{N} \sum_{i \in [N]} \frac{1}{N^{CF}} \sum_{j \in [N^{CF}]} \text{sim}_X(\mathbf{X}_{(i)}, \mathbf{X}^{CF}_{(i,j)}), \ \ \text{Proximity}_A = \frac{1}{N} \sum_{i \in [N]} \frac{1}{N^{CF}} \sum_{j \in [N^{CF}]} \text{sim}_A(\mathbf{A}_{(i)}, \mathbf{A}^{CF}_{(i,j)}),$$
(7)

where we use cosine similarity for $\text{sim}_X(\cdot)$, and accuracy for $\text{sim}_A(\cdot)$.

**Causality:** As it is difficult to obtain the true SCMs for real-world data, we focus on the causal relation of interest $R$ mentioned in dataset description. Similarly as [4], we measure the causality by reporting the ratio of counterfactuals which satisfy the causal constraints corresponding to $R$.

**Time:** the average time cost (seconds) of generating a counterfactual for a single graph instance.

### 4.4 Setup

We set the desired label $Y^*$ for each graph as its flipped label (e.g., if $Y = 0$, then $Y^* = 1$). For each graph, we generate three counterfactuals for it ($N^{CF} = 3$). Other setup details are in Appendix B.

### 4.5 RQ1: Performance of Different Methods

To evaluate our framework CLEAR, we compare its CFE generation performance against the state-of-the-art baselines. From the results in Table 1, we summarize the main observations as follows: 1) **Validity and proximity.** Our framework CLEAR achieves good performance in validity and proximity. This observation validates the effectiveness of our method in achieving the basic target of CFE generation. a) In validity, CLEAR obviously outperforms all baselines on most datasets. Random, EG-IST, and EG-RM perform the worst due to their random nature; GNNExplainer can only remove edges and nodes, which also limits its validity; CF-GNNExplainer and MEG perform well as their optimization is designed for CFE generation; b) In Proximity$_A$, CLEAR outperforms all non-random baselines. EG-RM performs the best in Proximity$_A$ because most graphs are very sparse, thus only removing edges can change the graph relatively less than other methods. As the baselines either cannot perturb node features, or their perturbation approach on node features cannot fit well in our setting, we do not compare Proximity$_X$ with them. 2) **Time.** CLEAR significantly outperforms all baselines in time efficiency. Most of the baselines generate CFEs in an iterative way, and MEG needs to enumerate all perturbations at each step. GNNExplainer and CF-GNNExplainer optimize on every single instance, which limits their generalization. All the above reasons erode their time efficiency. While in our framework, the generative mechanism enables efficient CFE generation and generalization on unseen graphs, thus brings substantial improvement in time efficiency. 3) **Causality.** CLEAR dramatically outperforms all baselines in causality. We contribute the superiority of our framework w.r.t. causality in two key factors: a) different from some baselines (e.g., GNNExplainer) optimized on each single graph, our framework can better capture the causal relations among different

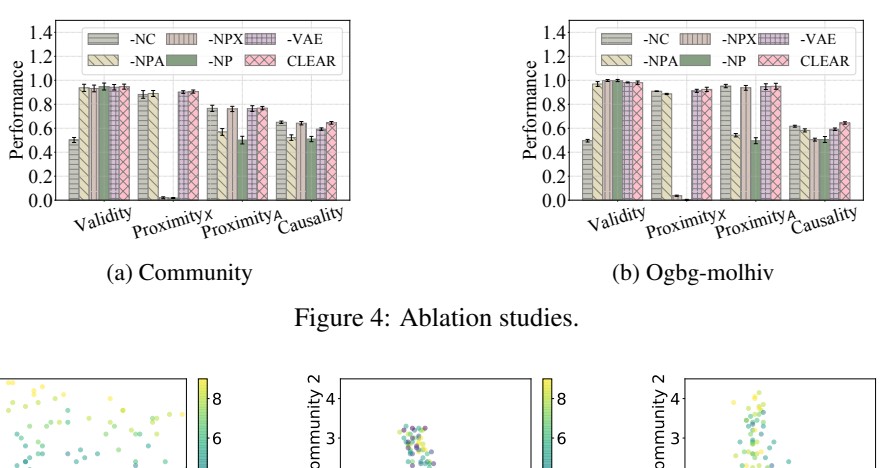

(a) Community                 (b) Ogbg-molhiv

Figure 4: Ablation studies.

(a) Original data       (b) CFEs from CLEAR-VAE       (c) CFEs from CLEAR

Figure 5: Explainability through CFEs on Community.

variables in data by leveraging the data distribution of the training set; b) our framework utilizes the auxiliary variable to better identify the underlying causal model and promote causality.

## 4.6 RQ2: Ablation Study

To evaluate the effectiveness of different components in CLEAR, we conduct ablation study with the following variants: 1) **CLEAR-NC**. In this variant, we remove the counterfactual prediction loss; 2) **CLEAR-NPA**, we remove the similarity loss w.r.t. graph structure; 3) **CLEAR-NPX**, we remove the similarity loss w.r.t. node features; 4) **CLEAR-NP**, we remove all the similarity loss; 5) **CLEAR-VAE**, the backbone of our framework. As shown in Fig. 4, we have the following observations: 1) The validity of CLEAR-NC degrades dramatically due to the lack of counterfactual prediction loss; 2) The performance w.r.t. proximity is worse in CLEAR-NPA, CLEAR-NPX, and CLEAR-NP as the similarity loss is removed. Besides, removing the similarity loss can also hurt the performance of causality when the variables in the causal relation of interest $R$ are involved. For example, in Community, CLEAR-NPA performs much worse in causality (as $R$ in Community involves node degree in graph structure), while in Ogbg-molhiv, the performance in causality of CLEAR-NPX is eroded (as $R$ on Ogbg-molhiv involves node features); 3) The performance w.r.t. causality is impeded in CLEAR-VAE. This observation validates the effectiveness of the auxiliary variable for promoting causality. Similar observations can also be found in the ablation study on the IMDB-M dataset, which is shown in Appendix C.

## 4.7 RQ3: Explainability through CFEs

To investigate how CFE on graphs promote model explainability, we take a closer look in the generated counterfactuals. Due to the space limit, we only show our investigation on the Community dataset. Fig. 5(a) shows the distribution of two variables: the average node degree in the first community and in the second community in the original dataset, i.e., $\deg_1(A)$ and $\deg_2(A)$. Fig. 5(b) shows the distribution of these two variables in counterfactuals generated by CLEAR-VAE. We observe that these counterfactuals are distributed close to the decision boundary, i.e., $\deg_1(A) = \mathrm{ADG}_1$, where $\mathrm{ADG}_1$ is a constant around 2. This is because that the counterfactuals are enforced to change their predicted labels with perturbation as slight as possible. Fig. 5(c) shows the distribution of these two variables $\deg_1(A)$ and $\deg_2(A)$ in counterfactuals generated by CLEAR. Different colors denote different values of the auxiliary variable $S$. Notice that based on the causal model (in Appendix B), the exogenous variables are distributed in a narrow range when the value of $S$ is fixed, thus the same color also indicates similar values of exogenous variables. We observe that compared with the

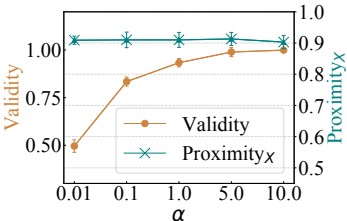 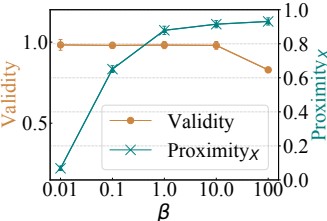

Figure 6: Parameter studies on Ogbg-molhiv.

color distribution in Fig. 5(b), the color distribution in Fig. 5(c) is more consistent with Fig. 5(a). This indicates that compared with CLEAR-VAE, CLEAR can better capture the values of exogenous variables, and thus the counterfactuals generated by CLEAR are more consistent with the underlying causal model. To better illustrate the explainability provided by CFE, we further conduct case studies to compare the original graphs and their counterfactuals in Appendix C.

### 4.8 RQ4: Parameter Study

To evaluate the robustness of our method, we test the model performance under different settings of hyperparameters. We vary $\alpha \in \{0.01, 0.1, 1.0, 5.0, 10.0\}$ and $\beta \in \{0.01, 0.1, 1.0, 10.0, 100\}$. Due to the space limit, we only show the parameter study on Ogbg-molhiv, but similar observations can be found in other datasets. As shown in Fig. 6, the selection of $\alpha$ and $\beta$ controls the tradeoff between different goals in the objective function. But generally speaking, the proposed framework is not very sensitive to the hyparameter setting. More studies regarding other parameters are in Appendix C.

## 5 Related Work

**Counterfactual explanations on tabular data.** Counterfactual explanations have attracted increasing attentions [2, 3, 5, 25]. Especially, recent works also consider more aspects in CFEs, such as actionability [26, 27], sparsity [28, 29, 30], data manifold closeness [31, 32], diversity [29, 33], feasibility [4, 27], causality [4, 19, 20], and amortized inference [4]. These methods include model-agnostic [27, 30, 32] methods and model-accessible [2, 25] methods. Recently, a few studies [4, 10] develop generative CFE generators based on variational autoencoder. However, most of current CFE approaches only focus on tabular or image datasets, and can not be directly grafted for the graph data.

**Counterfactual explanations on graphs.** There have been a few studies related to CFEs on graphs [6, 9, 10, 11, 12, 13]. Among them, GNNExplainer [12] identifies subgraphs which are important for graph neural network (GNN) model prediction [34, 35, 36, 37, 38, 39, 40]. It can be adapted to generate counterfactuals by perturbing the identified important subgraphs. Similarly, RCExplainer [10] generates CFEs by removing important edges from the original graph, but it is based on an assumption that GNN prediction model is partially accessible. CF-GNNExplainer [9] studies counterfactual explanations for GNN models in node classification tasks. Besides, a few domain-specific methods [6, 11] are particularly designed for certain domains such as chemistry and brain diagnosis. However, all the above methods are either limited in a specific setting (e.g., the prediction model is accessible), or heavily based on domain knowledge. Many general and important issues in model-agnostic CFE generation on graphs still lack exploration.

**Graph generative models.** Many efforts [16, 41, 42, 43] have been made in graph generative models recently. Among them, GraphVAE [16] develops a VAE-based mechanism to generate graphs from continuous embeddings. GraphRNN [41] is an autoregressive generative model for graphs. It generates graphs by decomposing the graph generation process into a sequence of node and edge formations. Furthermore, a surge of domain-specific graph generative models [44, 45, 46] are developed with domain knowledge incorporated. Although different from CFE methods in their goals, graph generative models can serve as the cornerstone of our work for CFE generation on graphs. Our framework can be compatible with techniques in different graph generative models.

## 6 Conclusion

In this paper, we study an important problem of counterfactual explanations on graphs. More specifically, we aim to facilitate the optimization, generalization, and causality in CFE generation

on graphs. To address this problem, we propose a novel framework CLEAR, which uses a graph variational autoencoder mechanism to enable efficient optimization in graph data, and generalization to unseen graphs. Furthermore, we promote the causality in counterfactuals by improving the model identifiability with the help of an auxiliary observed variable. Extensive experiments are conducted to validate the superiority of the proposed framework in different aspects. In the future, more properties of the counterfactuals in graphs, such as diversity, data manifold closeness can be considered. Besides, incorporating different amount and types of prior knowledge regarding the causal models into CFE generation on graphs is also an interesting direction.

## Acknowledgements

This work is supported by the National Science Foundation under grants IIS-2006844, IIS-2144209, IIS-2223769, IIS-2106913, IIS-2008208, IIS-1955151, CNS-2154962, BCS-2228534, the JP Morgan Chase Faculty Research Award, the Cisco Faculty Research Award, and the 3 Cavaliers seed grant. Any opinions, findings, and conclusions or recommendations expressed in this material are those of the author(s) and do not necessarily reflect the views of the funding agencies.

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
