## A Theory

**Theorem 1.** The evidence lower bound (ELBO) to optimize the framework is:

$$\ln P(G^{CF}|S,Y^*,G) \geq \mathbb{E}_Q[\ln P(G^{CF}|Z,S,Y^*,G)] - \mathrm{KL}(Q(Z|G,S,Y^*)\|P(Z|G,S,Y^*)), \tag{8}$$

*Proof.*

$$
\begin{aligned}
&\ln P(G^{CF}|S,Y^*,G) \\
&= \ln \int_Z P(G^{CF},Z|S,Y^*,G)dZ \\
&= \ln \int_Z Q(Z|G,S,Y^*)\frac{P(G^{CF},Z|S,Y^*,G)}{Q(Z|G,S,Y^*)}dZ \\
&\geq \int_Z Q(Z|G,S,Y^*)\ln \frac{P(G^{CF},Z|S,Y^*,G)}{Q(Z|G,S,Y^*)}dZ \\
&= \mathbb{E}_Q[\ln \frac{P(G^{CF},Z|S,Y^*,G)}{Q(Z|G,S,Y^*)}] \\
&= \mathbb{E}_Q[\ln \frac{P(G^{CF}|Z,S,Y^*,G)\cdot P(Z|G,S,Y^*)}{Q(Z|G,S,Y^*)}] \\
&= \mathbb{E}_Q[\ln P(G^{CF}|Z,S,Y^*,G)] - \mathbb{E}_Q[\ln \frac{Q(Z|G,S,Y^*)}{P(Z|G,S,Y^*)}] \\
&= \mathbb{E}_Q[\ln P(G^{CF}|Z,S,Y^*,G)] - \mathrm{KL}(Q(Z|G,S,Y^*)\|P(Z|G,S,Y^*)).
\end{aligned}
\tag{9}
$$

$\square$

## B Reproducibility

In this section, we provide more details of model implementation and experiment setup for reproducibility of the experimental results.

### B.1 Details of Model Implementation

### B.1.1 Details of the Prediction Model

The prediction model $f$ is implemented with a graph neural network based model. Specifically, this prediction model includes the following components:

- Three layers of graph convolutional network (GCN) [34] with learnable node masks.
- Two graph pooling layers with mean pooling and max pooling, respectively.
- A two-layer multilayer perceptron (MLP) with batch normalization and ReLU activation function.

The prediction model uses negative log likelihood loss. The representation dimension is set as 32. We use Adam optimizer, set the learning rate as $0.001$, weight decay as $1e-5$, the training epochs as 600, dropout rate as $0.1$, and batch size as 500. As shown in Table 2, we observe that the prediction model $f$ achieves high performance of graph classification on all datasets.

Table 2: Performance of the prediction model on the test data of the three datasets.

| Dataset | Community | Ogbg-molhiv | IMDB-M |
|---|---|---|---|
| Accuracy | $0.949 \pm 0.006$ | $0.897 \pm 0.004$ | $0.995 \pm 0.002$ |
| AUC-ROC | $0.993 \pm 0.002$ | $0.997 \pm 0.002$ | $1.000 \pm 0.001$ |
| F1-score | $0.947 \pm 0.005$ | $0.906 \pm 0.004$ | $0.994 \pm 0.003$ |

### B.1.2 Details of CLEAR

CLEAR is designed in a general way, which can be adaptable to different graph representation learning modules and different techniques in graph generative models. Specifically, in our implementation, we apply a graph convolution [34] based module as the encoder, and use a multilayer perceptron (MLP) as the decoder. We also use MLPs to learn the mean and covariance of the prior distribution of the latent variables. We choose the pairwise distance using $L_2$ norm to implement $d_X(\cdot)$, and use cross entropy loss to implement $d_A(\cdot)$. We implement the counterfactual prediction loss with the negative log likelihood loss. Following [16], we assume that the maximum number of nodes in the graph is $k$, and use a graph matching technique to align the input graph and counterfactuals. The detailed implementation contains the following components:

- **Prior distribution:** Two different two-layer MLPs are used to learn the mean and covariance of the prior distribution $P(Z|G, S, Y^*)$, respectively.

- **Encoder:** The encoder contains a single-layer graph convolutional network, a graph pooling layer with mean pooling, and two linear layers with batch normalization and ReLU activation function to learn the mean and covariance of the approximate posterior distribution $Q(Z|G, S, Y^*)$.

- **Decoder:** The decoder uses two three-layer MLPs to output the node features and graph structure of the counterfactual $G^{CF}$, respectively. These MLPs use batch normalization, and take ReLU as activation function in the middle layers. At the last layer of decoder, the MLP which generates the graph structure uses Sigmoid as activation function to output a probabilistic adjacency matrix $\hat{A}^{CF}$ with elements in range $[0, 1]$.

Inspired by [16], we use a graph matching technique to align the input graph and counterfactuals. Specifically, we learn a graph matching matrix $M = \{0, 1\}^{k \times n}$ to match the generated counterfactual with the original explainee graph. Here, $n$ is the number of nodes in the original graph, and $M_{(i,j)} = 1$ if and only if node $i$ is in $G^{CF}$ and node $j$ is in $G$, and $M_{(i,j)} = 0$ otherwise.

## B.2 Details of Experiment Setup

### B.2.1 Baseline Settings

Here we introduce more details of baseline setting:

- **Random**: For each explainee graph, it randomly perturbs the graph structure for at most $T = 150$ steps. In each step, at most one edge can be inserted or removed. We stop the process if the perturbed graph can achieve a desired predicted label.

- **GNNExplainer:** For each graph, GNNExplainer [12] outputs an edge mask which estimates the importance of different edges in model prediction. In CFE generation, we set a threshold 0.5 and remove edges with edge mask weight smaller than the threshold. Although GNNExplainer can also identify important node features in a similar way, when we apply GNNExplainer for CFE generation, the perturbation on node features cannot be designed as straightforwardly as the perturbation on graph structure, thus we did not involve perturbation on node features in GNNExplainer.

- **CF-GNNExplainer:** CF-GNNExplainer [9] is originally proposed for node classification tasks, and it only focuses on the perturbations on the graph structure. Originally, for each explainee node, it takes its neighborhood subgraph as input. To apply it on graph classification tasks, we use the graph instance as the neighborhood subgraph, and assign the graph label as the label for all nodes in the graph. We set the number of iterations to generate counterfactuals for each graph as 150.

- **MEG:** MEG [6] is specifically proposed for molecular prediction tasks. This model explicitly incorporates domain knowledge in chemistry. The CFE generator is developed based on reinforcement learning, and it designs the reward based on the prediction on the counterfactual, as well as the similarity between the original graph and the counterfactual. In each step, MEG enumerates all possible perturbations (e.g., adding an atom) which are valid w.r.t. chemistry rules to form an action set. We apply it to general graphs by removing the constraints of domain knowledge, and enumerating the perturbations as: 1) adding or removing a node; 2) adding or removing an edge; 3) staying the same. We set the number of action steps as 150.

Table 3: Detailed statistics of the datasets.

| Dataset | Community | Ogbg-molhiv | IMDB-M |
|---|---|---|---|
| # of graphs | $10,000$ | $31,957$ | $1,160$ |
| Avg # of nodes | 20 | 20.8 | 9.4 |
| Avg # of edges | 45.0 | 22.4 | 32.8 |
| Max # of nodes | 20 | 30 | 15 |
| # of classes | 2 | 2 | 2 |
| Feature dimension | 16 | 11 | 2 |
| Avg node degree | 2.24 | 1.07 | 3.4 |

### B.2.2 Datasets

For each dataset, we filter out the graphs with the number of nodes larger than a threshold $k$. The setting of $k$ (i.e., max # of nodes) can be found in Table 3. As some of the baselines need to be optimized separately for each graph, we compare the performance of all methods on a small set of test data with 20 graphs for evaluation in RQ1. For other RQs, we evaluate our framework on the whole test data.

**1. Community.** We first generate a synthetic dataset in which we can fully control the data generation process. In this dataset, each graph consists of two 10-node communities generated using the Erdös-Rényi (E-R) model [24] with edge rate $p_1$ and $p_2$, respectively. Specifically, we simulate the data with the following causal model:

$$S \sim \text{Uniform}(\{0, ..., 9\}), \ p_1 = U_1 \sim \text{Uniform}([0, 1]),$$

$$U_2 \sim \text{Uniform}([\delta S + b, \delta(S+1) + b]), \ p_2 = \max\{0, \min\{1, -0.15p_1 + U_2\}\},$$

$$X \sim \mathcal{N}(0, I), \ Y \sim \text{Bernoulli}(\text{Sigmoid}(\deg_1(A) - \text{ADG}_1 + \epsilon_y)). \quad (10)$$

$U_1$ and $U_2$ are two exogenous variables associated with $p_1$ and $p_2$, respectively. Notice that $p_2$ is determined by $p_1$ and $U_2$. Here, the auxiliary variable $S$ provides help to infer the value of exogenous variables (specifically, $U_2$ in this case). We set $\delta = 0.085, b = 0.15$. $p_1$ and $p_2$ thereby generate the graph structure inside the two communities, respectively. We also randomly add few edges between these two communities. The edges connecting two communties are randomly generated with an edge rate of $0.05$. In this way, the adjacency matrix $A$ of each graph is simulated. $\deg_1(A)$ (determined by $p_1$) denotes the average node degree in the first community of each graph $A$. **Label generation:** The label $Y$ is determined by $\deg_1(A)$ together with a Gaussian noise $\epsilon_y \sim \mathcal{N}(0, 0.01^2)$. $\text{ADG}_1$ is a constant, which is the average value of $\deg_1(A)$ over all graphs. **Causality:** To elicit a different predicted label, $\deg_1(A)$ in the counterfactual is supposed to be perturbed, while other variables can remain the same. But considering that with the above causal model, when $\deg_1(A)$ increases (decreases), the average node degree in the second community $\deg_2(A)$ (determined by $p_2$) should decrease (increase) correspondingly. We take this causal relation $\deg_1(A) \to \deg_2(A)$ as our causal relation of interest, and denote it as $R$. Correspondingly, we define a causal constraint for evaluation of causality: "$(\deg_1(A^{CF}) > \deg_1(A)) \Rightarrow (\deg_2(A^{CF}) < \deg_2(A))$" OR "$(\deg_1(A^{CF}) < \deg_1(A)) \Rightarrow (\deg_2(A^{CF}) > \deg_2(A))$".

**2. Ogbg-molhiv.** Ogbg-molhiv is adopted from the MoleculeNet [47] datasets. All molecules are preprocessed with RDKit [48]. The original node features are 9-dimensional, containing atom features such as atomic number, formal charge and chirality. In this dataset, each graph stands for a molecule, where each node represents an atom, and each edge is a chemical bond. As the ground-truth causal model is unavailable, we simulate the label and causal relation of interest as follows: **Label generation:** $Y \sim \text{Bernoulli}(\text{Sigmoid}(X_1 - \text{AVG}_{x1}))$, where $X_1$ is the average value of a synthetic node feature over all nodes in each graph. This node feature is generated for each node from distribution $\text{Uniform}(0, 1)$. $\text{AVG}_{x1}$ means the average value of $X_1$ over all graphs. **Causality:** We also add a causal relation of interest $R$ between $X_1$ and another synthetic node feature $X_2$: $X_2 = U_2 + 0.5X_1$. Here $U_2$ is simulated in a similar way as the Community dataset. Correspondingly, we have the following causal constraint: "$(X_1^{CF} > X_1) \Rightarrow (X_2^{CF} > X_2)$" OR "$(X_1^{CF} < X_1) \Rightarrow (X_2^{CF} < X_2)$".

**3. IMDB-M.** This dataset contains movie collaboration networks from IMDB. In each graph, each node represents an actor or an actress, and each edge represents the collaboration between two actors or

actresses in the same movie. Similarly as the above datasets, we simulate the label and causal relation of interest as follows: **Label generation:** $Y \sim \text{Bernoulli}(\text{Sigmoid}(\deg(A) - \text{ADG} + \epsilon_y))$. $\deg(A)$ is the average node degree in graph with adjacency matrix $A$. ADG is the average value of $\deg(A)$ over all graphs. **Causality:** We also add a causal relation of interest $R$ from the average node degree to a synthetic node feature: $X_1 = U_1 + 0.5\deg(A)/\text{ADG}$, where $U_1 \sim \text{Uniform}[0.1S, 0.1S + 0.1]$, $S \sim \text{Uniform}\{0, ..., 9\}$. We denote the causal relation $\deg(A) \rightarrow X_1$ as $R$, and define an associated causal constraint: "$(\deg(A^{CF}) > \deg(A)) \Rightarrow (X_1^{CF} > X_1)$" OR "$(\deg(A^{CF}) < \deg_1(A)) \Rightarrow (X_1^{CF} < X_1)$".

### B.2.3 Experiment Settings

All the experiments are conducted in the following environment:

- Python 3.6
- Pytorch 1.10.1
- Pytorch-geometric 1.7.0
- Scikit-learn 1.0.1
- Scipy 1.6.2
- Networkx 2.5.1
- Numpy 1.19.2
- Cuda 10.1

In all the experiments of counterfactual explanation, each dataset is randomly split into 60%/20%/20% training/validation/test set. Unless otherwise specified, we set the hyperparameters as $\alpha = 5.0$ and $\beta = 10.0$. The batch size is 500, and the representation dimension is 32. The graph prediction models trained on all the above datasets perform well in label prediction (AUC-ROC score over 95% and F1 score over 90% on test data). We use NetworkX [49] to generate synthetic graphs. In our CFE generator CLEAR, the learning rate is 0.001, the number of epochs is 1,000. All the experimental results are averaged over ten repeated executions. The implementation is based on Pytorch. We use the Adam optimizer for model optimization.

## C More Experimental Results

### C.1 Ablation Study

Fig. 7 shows the results of ablation studies on the IMDB-M dataset. The observations are generally consistent with the observations on other two datasets as described in Section 4.6.

### C.2 Case Study

To better illustrate the explainability provided by CFE, we further conduct case studies to compare the original graphs and their counterfactuals. In the Community dataset, Fig. 8 shows the change from original graphs to their counterfactuals w.r.t. the average node degree in the first community and in the second community, i.e., $\deg_1(A)$ and $\deg_2(A)$. Here, Fig. 8 has the same x-axis and y-axis as Fig. 5. In Fig. 8, we randomly select 6 graphs and show them in different shapes of markers. The colors denote their values of $S$ with the same colorbar in Fig. 5(a-c). In Fig. 8, we connect the pairs (original, counterfactual generated by CLEAR) with solid lines, and connect the pairs (original, counterfactual generated by CLEAR-VAE) with dashed lines. We have the following observations: 1) Compared with the input graph, the counterfactuals generated by CLEAR-VAE and CLEAR both make the correct perturbations to achieve the desired label (moving the variable $\deg_1(A)$ across the decision boundary at around $\deg_1(A) = 2$); 2) The counterfactuals generated by CLEAR better match the causality than CLEAR-VAE in two aspects: a) Qualitatively, the counterfactuals generated by CLEAR better satisfy the causal constraints introduced in the dataset description, i.e., $\deg_2(A)$ increases (decreases) when $\deg_1(A)$ decreases (increases); 2) Quantitatively, the changes from original graphs to their counterfactuals fit in well with the associated structural equations $(\deg_1(A), U_2) \rightarrow \deg_2(A)$. Notice that in counterfactuals, $\deg_1(A)$ changes but $U_2$ is supposed to maintain its original value.

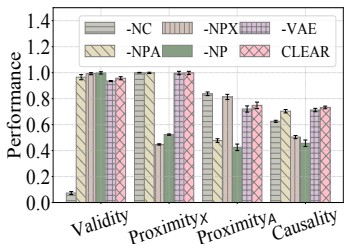

Figure 7: Ablation studies on the IMDB-M dataset.

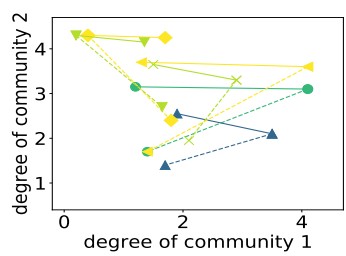

Figure 8: Case study.

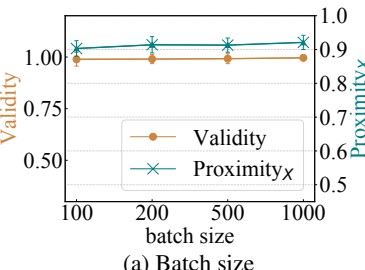

(a) Batch size

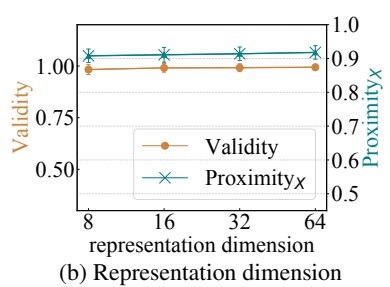

(b) Representation dimension

Figure 9: Parameter studies on Ogbg-molhiv regarding batch size and representation dimension.

## C.3  Parameter Study

Here, we conduct further parameter study with respect to the batch size and representation dimension. Specifically, we vary the batch size from range $\{100, 500, 1000, 2000\}$, and the representation dimension from range $\{8, 16, 32, 64\}$. From the results shown in Fig. 9, we observe that the performance of CLEAR under different settings of these parameters is generally stable. This observation further validates the robustness of our framework.

## D  Further Discussion

**CFEs in Other Tasks on Graphs.** In this paper, we mainly focus on the task of graph classification, but it is worth noting that the proposed framework CLEAR can also be used for counterfactual explanations in other tasks such as node classification. More specifically, in a node classification task, CLEAR can generate CFEs for nodes with the same loss function in Eq. (5). But differently, the encoder here learns node representations instead of graph representations at the bottleneck layer. Besides, in this case, $Y^*$ is a vector which contains the desired labels for all the training nodes on graph $G$, and $S$ is the vector of auxiliary variables for all the training nodes. Notice that in a graph, nodes are often not independent. To obtain a valid counterfactual for an explainee node, not only can we change the explainee node's own features and adjacent edges, but we can also change other nodes' features or any other part of the graph structure. Therefore, the decoder still needs to generate a graph $G^{CF}$ as a counterfactual (but this process can be more efficient, as in many scenarios, we only need to generate counterfactuals for each node's neighboring subgraph instead of the whole graph). Similarly, our framework can also be extended to generate CFEs for graphs in other tasks, such as link prediction.

**Limitation, Future Work, and Negative Societal Impacts.** In this work, we mainly focus on promoting optimization, generalization, and causality in counterfactual explanations on graphs, while other important targets (e.g., actionability [26], sparsity [28], diversity [29], and data manifold closeness [31]) in traditional counterfactual explanations could be considered in graph data in the future. Noticeably, the definition and evaluation metrics with respect to these targets should be specifically tailored for graphs, rather than directly employed in the same way as other types of data. Besides, in terms of causality, another interesting direction is incorporating different levels of prior knowledge and assumptions regarding the underlying causal model into CFE generation on graphs, and quantifying the influence of different levels of prior knowledge and assumptions on the CFE performance. Currently, we have not found any negative societal impact regarding this work.