# OpenReview forum: "CLEAR: Generative Counterfactual Explanations on Graphs"
_NeurIPS.cc/2022/Conference — NeurIPS 2022 Accept_

### Official Review · Reviewer_SccR · 2022-06-29

**Rating:** 6
**Confidence:** 5
**Soundness:** 4 excellent
**Presentation:** 3 good
**Contribution:** 4 excellent

**Summary:**

The authors present a counterfactual generation method for graphs. The proposed method (CLEAR) aims to perturb a given graph in a way that is still close to the original while changing a classifier prediction and respecting the causality in the data generating process. This is achieved by training a graph VAE conditioned on the target label and an auxiliary variable $S$ used by the original data generating process (an SCM), which helps to identify the causal model. The overall loss encourages counterfactuals to be close (wrt a distance metric) to the original graph while achieving the desired label ($Y^*$) and inferring the correct latent structure by keeping the latent codes $Z$ close to a distribution that is conditioned on $S$ and $Y^*$. In experiments they show that the proposed method achieves a higher number of valid counterfactuals that are more proximal while keeping causal relations than previous state of the art. They also provide an ablation study on the different parts of their model.

**Questions:**

* Could you better motivate the counterfactual explanation problem in the introduction and the abstract? The idea is that the reader gets the "big picture" before you focus on the concrete problem of improving graph counterfactual methods.
* Could you verify whether your model is or not producing the same counterfactual 3 times? (if it does it is ok, but I think this is important information for the reader to have)
* Could you clarify what causality you want to preserve? (see clarity above)
* Why did you skip 0.1 in the hyperparameter sweep? (see above)

**Limitations:**

Yes, the authors include a section in the appendix.

**Strengths And Weaknesses:**

Overall Review
===========
Overall, this paper tackles an important problem, the method is sound, and the results are encouraging. Some points could be improved, such as motivating the need of graph counterfactuals, or checking whether the model is repeating the same counterfactual three times. Therefore I recommend "Weak Accept" and I would be happy to raise the score based on the rebuttal and the other reviews.

Strengths
=======
* The problem of understanding the behavior of ML systems is an important one.
* The proposed method is sound and it performs better compared to previous state of the art.
* The authors provide the code, ablations, and information for reproducibility.

Weaknesses
=========
* This work focuses strongly on the method and comparison with previous state of the art but it lacks a bit of perspective on what is the final purpose of this research. Some more motivation in the introduction and some qualitative results showing interesting findings on a real graph would make this work more appealing. I like the example in Section 2, so you can focus more on motivating the problem in general rather than giving more examples in the introduction.
* In the ablation study there is no clear difference between VAE and CLEAR. (The difference is, however, more clear on Figure 4).
* Since the proposed method predicts 3 counterfactuals, it could just predict three times the same (or very similar) graph. This would improve the validity metric while not adding any additional value to the user. Several recent works focus on providing a set of diverse [28, A, B] and non-trivial explanations [C] (unexpected failure modes of the model), that are more useful for the end user. In fact, in Appendix D, the authors acknowledge that optimizing for diversity is a current limitation, but it would be interesting to study whether the VAE is repeating the same graph or producing different graphs.

[28] Mothilal, Ramaravind K., Amit Sharma, and Chenhao Tan. "Explaining machine learning classifiers through diverse counterfactual explanations." Proceedings of the 2020 conference on fairness, accountability, and transparency. 2020.

[A] Smyth, Barry, and Mark T. Keane. "A Few Good Counterfactuals: Generating Interpretable, Plausible and Diverse Counterfactual Explanations." arXiv preprint arXiv:2101.09056 (2021).

[B] Hvilshøj, Frederik, Alexandros Iosifidis, and Ira Assent. "On Quantitative Evaluations of Counterfactuals." arXiv preprint arXiv:2111.00177 (2021).

[C] Rodríguez, Pau, et al. "Beyond trivial counterfactual explanations with diverse valuable explanations." Proceedings of the IEEE/CVF International Conference on Computer Vision. 2021.



Detailed comments
===============
Originality
-----------
The proposed framework, and particularly the focus on causality preservation when generating counterfactuals, are novel to the best of my knowledge.

Quality
--------
The technical and written quality are good, assumptions are clearly stated, and proofs are provided in the Appendix.

Clarity
-------
Overall, the text is well-written and easy to follow. In section 4.7 I believe that references to Figure 8 should be to Figure 4. In Section 4.8 it is not clear why you do not test $0.1$. Many times you talk about preserving causality and, without any context, it is not clear if you mean that you are reconstructing a causal graph (with directed edges) and you want to preserve edge directions, or if you want the generated graphs to be compatible with the original SCM. After some time reading it becomes clear but it would be better if you made it more clear since the beginning.

Significance
--------------
Given the increasing impact of machine learning applications in our lives, it is important to better understand how models make their predictions. However, the text does not fully transmit the significance of this work.

Reproducibility
-----------------
The authors provide code, equations, and the necessary assumptions to reproduce their results.

---

> ### Author Response · Authors · 2022-08-02
> **To Reviewer SccR**
>
> Thank you for carefully reviewing our paper and these comments. We offer the clarifications below to solve your concerns.
>
> ### Q1: Motivating the problem in general rather than giving more examples in abstract and introduction .
>
> Thank you for this suggestion. We’ve added more general motivation in abstract and introduction in the new version (highlighted in blue).
>
> “Generally, CFE promotes human interpretation through the comparison between the explainee instance $X$ with predicted label $Y$ and its counterfactual $X^′$ with predicted label $Y' \neq Y$. With its intuitive nature, CFEs can be deployed in various scenarios such as loan application and legal framework [1]. Different from traditional CFE studies on tabular or image data, recently, CFE on graphs is also an emerging field with applications in many graph structure related domains such as molecular analysis and career networking.”
>
> ### Q2: In the ablation study there is no clear difference between VAE and CLEAR.
>
> Compared with CLEAR-VAE, CLEAR improves the causality score. For other metrics, they are not expected to have a clear difference, because CLEAR differs from CLEAR-VAE in its ability of considering causality.
>
> ### Q3: Could you verify whether your model is or not producing the same counterfactual 3 times?
>
> Thanks for this helpful comment. Although we did not directly encourage diversity in this work, the sampling process in our VAE-based counterfactual explanation generator decreases the probability of generating counterfactuals which are exactly the same. Actually, in our experiments, we’ve never observed counterfactuals which are exactly the same when $N^{CF}=3$. But it would also be an interesting future work to explicitly consider diversity into counterfactual explanations on graphs.
>
> ### Q4: Could you clarify what causality you want to preserve?
>
> We clarify that the causality in this paper means that we aim to generate counterfactual graphs which are compatible with the original structural causal model (SCM).
>
> ### Q5: Why did you skip 0.1 in the hyperparameter sweep?
>
> We have updated the results of 0.1 in Fig. 6 of the new version. The main observation of hyperparameter study is that the performance of validity and proximity of CLEAR is generally good unless $\alpha$ and $\beta$ are too unbalanced. This observation also holds when we set $\alpha$ or $\beta$ as 0.1.
>
> ### Q6: Typo in references to Figure 8
>
> Thank you for carefully reading our manuscript, we have revised it in the new version.
>
> [1] Verma, Sahil, John Dickerson, and Keegan Hines. "Counterfactual explanations for machine learning: A review." arXiv preprint arXiv:2010.10596 (2020).

---

> > ### Author Response · Authors · 2022-08-07
> > **Looking forward to your further feedback**
> >
> > Dear Reviewer SccR,
> >
> > Thank you again for your valued comments! We have responded to your initial questions, and we are looking forward to your further feedback. We will be happy to answer any further questions you may have.
> >
> > Thank you.
> >
> > Authors

---

> > > ### Comment · Reviewer_SccR · 2022-08-09
> > > **Thank you**
> > >
> > > Your responses clarify most of my doubts. The only part I am still concerned about is that sampling does not guarantee that your model does produce multiple counterfactuals that are different from each other and thus, it is not clear if the evaluation can be trusted.

---

> > > > ### Author Response · Authors · 2022-08-09
> > > > **Respond to Reviewer SccR**
> > > >
> > > > Thank you for your additional feedback! Although encouraging diversity is not our main focus in this paper, the probability of generating the same counterfactual from different times of sampling is very low due to the complexity of graph structure. In our current experiments, over 99% counterfactuals for the same graph are not the same. Besides, it is very easy to explicitly involve further constraints such as diversity in our framework, depending on the application scenario.
> > > >
> > > > Please let us know if you have further comments or concerns, thanks!

---

> > > > > ### Comment · Reviewer_SccR · 2022-08-09
> > > > > **Response**
> > > > >
> > > > > When you say "not the same" you mean that the discrete graph structure is different? If so, that is encouraging, you might want to clarify this in the paper.

---

> > > > > > ### Author Response · Authors · 2022-08-09
> > > > > > **Response**
> > > > > >
> > > > > > Yes. The graph structure of the counterfactuals is different. Thank you for this suggestion, we will add this clarification in the paper.

---

### Official Review · Reviewer_Htsg · 2022-07-05

**Rating:** 6
**Confidence:** 5
**Soundness:** 3 good
**Presentation:** 3 good
**Contribution:** 3 good

**Summary:**

In this work, the authors study the problem of generating counterfactual explanations for graphs using Graph Neural Networks (GNN). In contrast to some existing studies, the work lists three unaddressed counterfactual properties, i.e., i) discrete optimization of graphs, ii) generalization of counterfactuals on unseen graphs, and iii) ensuring causality in the generated counterfactuals without prior knowledge of the causal model. The work leverages a graph variational autoencoder-based framework to propose CLEAR (generative CounterfactuaL ExplAnation geneRator for graphs) that ensures the optimization and generalization of discrete counterfactual explanations for graphs and enforces causality by using an auxiliary variable for estimating the underlying causal model better.

**Questions:**

Please refer to the Strengths and Weaknesses section for questions.

**Limitations:**

Yes.

**Strengths And Weaknesses:**

Strengths:
1. The paper nicely enumerates three important desiderata for counterfactual explanation generators for graphs and describes their utilities with respect to counterfactual explanations.
2. The formulation of the objective for the CLEAR framework is clearly explained with proper derivations and descriptions of the individual components.
3. Extensive experiments with both synthetic and real-world graph datasets highlight the effectiveness of the CLEAR framework as compared to the baselines and show the utility of its components.

Weaknesses and Questions:
1. One of the major drawbacks of the work is that they don't detail the perturbation mechanism used for generating counterfactuals. They mention that: "CLEAR aims to generate counterfactuals with slight perturbations on the explainee graph to elicit a desired predicted label" but does not describe the perturbation process. This is crucial for understanding the framework as one of the main challenges in evaluating the reliability of generated counterfactual explanations is efficiently perturbing the input data. The perturbation process for generating similar graphs using very small perturbation is unclear, i.e., how to generate perturbed instances that are not out-of-distribution samples.

2. It is unclear from Section 4.7 and Appendix C.2 how the generated counterfactuals using CLEAR promote model explainability. The observation that the framework makes correct perturbation to achieve the target label by visualizing the degrees across the decision boundary is intuitive and shown in multiple previous works.

3. The proposed CLEAR framework uses a generative backbone CLEAR-VAE and claims that it can generate counterfactual explanations for unseen graphs. However, it should still require graphs to belong to the same distribution as the training data.

4. The paper details the problem of optimizing counterfactual explanation on graphs due to its discrete nature but then follows the optimization trick used by previous works for generating a counterfactual adjacency matrix.

5. The use of GNNExplainer as a baseline is unclear. The author assigns the given graph label to all the nodes inside the graph. It would be great if the authors explain this a bit more. Are they assigning the same label to all nodes and then generating explanations for a given node? Or are they aggregating all node explanations to generate a graph-level explanation?

6. It would be great if the authors can motivate using causality metric for comparison. It feels that the metric is biased towards the proposed framework as the auxiliary variable can provide additional information to identify the exogenous variables in the structural causal model, which is captured by the CLEAR-VAE training process.

---

> ### Author Response · Authors · 2022-08-02
> **To Reviewer Htsg**
>
> Thank you for the comments. We offer the clarifications below to solve your concerns.
>
> ### Q1: The perturbation process for generating similar graphs using very small perturbation is unclear, i.e., how to generate perturbed instances that are not out-of-distribution samples.
>
> The perturbation mechanism in our method is inferred by the proposed graph autoencoder in CLEAR, which is trained to optimize the objective (Eq. (2)). The loss function in Eq. (2) enforces CLEAR to learn a way to perturb the input graph and generate counterfactuals which 1) are similar to the input graph (enforced by the term $d(G, G^{CF})$ in Eq. (2), and 2) achieve the desired prediction (enforced by the term $l(f(G^{CF}), Y^∗)$). In our framework, the VAE backbone is optimized based on the evidence lower bound (ELBO), which is equivalent to maximizing a lower bound of the log likelihood of training data. This helps prevent generating out-of-distribution graphs.
>
> ### Q2: It is unclear from Section 4.7 and Appendix C.2 how the generated counterfactuals using CLEAR promote model explainability.
>
> Visualizing the perturbations is a commonly-used way for showing the ability of promoting model explanability for  counterfactual explanations [1], so we also follow this way in Section 4.7 and Appendix C.2. The results show that CLEAR can promote model explainability in two aspects: 1) CLEAR does find the decision boundary of the prediction model, and thus can explain how we can make perturbations to achieve a desired prediction; 2) the generated counterfactuals are consistent with the underlying causal model, therefore, CLEAR can provide more realistic perturbations for human interpretation.
>
> Before these results, the superiority of our method over existing methods in graph counterfactual explanation has already been validated in Table 1 by comparison in different metrics (e.g., validity, similarity, causality, etc.), so the good performance of CLEAR in these metrics has shown its advantage in promoting model explainability.
>
> ### Q3: The proposed CLEAR framework uses a generative backbone CLEAR-VAE and claims that it can generate counterfactual explanations for unseen graphs. However, it should still require graphs to belong to the same distribution as the training data.
>
> We should clarify that the word “generalization” in this paper is different from “domain generalization” for graphs from different distributions. In this paper, the point is that our method can be directly applied for unseen graphs in an inductive way without retraining the model, while most existing graph CFE methods either are based on enumeration, or need to be trained separately for each input graph.
>
> ### Q4: The paper details the problem of optimizing counterfactual explanation on graphs due to its discrete nature but then follows the optimization trick used by previous works for generating a counterfactual adjacency matrix.
>
> The discrete nature of graph structure brings difficulty for optimization, and thus many existing methods of counterfactual explanation on graphs are designed in an enumeration way. To circumvent this challenge, we allow gradient-based counterfactual explanation on graphs by 1) using a graph neural network (GNN) to handle the input graph, and 2) generating a continuous adjacency matrix and then mapping it to discrete values.
>
> As far as we know, our work is the first gradient-based optimization method for counterfactual explanation on graphs without additional assumption of the prediction model and application domain.
> Although gradient-based optimization has been employed in previous work of graph generation, in the area of counterfactual explanation on graphs, most existing methods are still in an enumeration way.
>
> For those few graph CFE methods which enable gradient-based optimization, most of them rely on specific domain knowledge [1] (e.g., chemical rules) or assumptions [2] about the prediction model (e.g., the prediction model is a graph neural network (GNN) model and its gradients or representations are accessible) to prune the search space or facilitate the optimization. However, these domain knowledge and assumptions limit their application in different scenarios, while our proposed method can optimize without these knowledge or assumptions.

---

> > ### Author Response · Authors · 2022-08-02
> > **To Reviewer Htsg**
> >
> > ### Q5: The use of GNNExplainer as a baseline is unclear. Are they assigning the same label to all nodes and then generating explanations for a given node? Or are they aggregating all node explanations to generate a graph-level explanation?
> >
> > We suppose the reviewer refers to CF-GNNExplainer. The original CF-GNNExplainer focuses on node classification, the input is a specific node’s surrounding subgraph, and the output is a perturbed subgraph to change the prediction of this node. We adapt CF-GNNExplainer for graph classification by taking the whole graph (instead of the subgraph of any specific node) as input, and optimizing the model until the graph classification (instead of node classification) label has been changed to the desired one. In this way, we are not optimizing for a specific node, and we do not need aggregation over all node explanations.
> > We’ve revised the baseline description for better clarification in the new version (highlighted in blue).
> >
> > ### Q6: It would be great if the authors can motivate using causality metric for comparison. It feels that the metric is biased towards the proposed framework as the auxiliary variable can provide additional information to identify the exogenous variables in the structural causal model, which is captured by the CLEAR-VAE training process.
> >
> > As far as we know, this is the first work which incorporates causality into counterfactual generation on graphs. We use the similar causality metrics in previous work of counterfactual explanation on i.i.d. data [3].
> >
> > [1] Numeroso, Danilo, and Davide Bacciu. "Meg: Generating molecular counterfactual explanations for deep graph networks." 2021 International Joint Conference on Neural Networks (IJCNN). IEEE, 2021.
> >
> > [2] Bajaj, Mohit, et al. "Robust counterfactual explanations on graph neural networks." Advances in Neural Information Processing Systems 34 (2021): 5644-5655.
> >
> > [3] Mahajan, Divyat, Chenhao Tan, and Amit Sharma. "Preserving causal constraints in counterfactual explanations for machine learning classifiers." arXiv preprint arXiv:1912.03277 (2019).

---

> > > ### Author Response · Authors · 2022-08-07
> > > **Looking forward to your further feedback**
> > >
> > > Dear Reviewer Htsg,
> > >
> > > Thank you again for your valued comments! We have responded to your initial questions, and we are looking forward to your further feedback. We will be happy to answer any further questions you may have.
> > >
> > > Thank you.
> > >
> > > Authors

---

> > > > ### Comment · Reviewer_Htsg · 2022-08-08
> > > > **Rebuttals Response**
> > > >
> > > > Thank you for the detailed response. I am happy to increase my score.

---

> > > > > ### Author Response · Authors · 2022-08-08
> > > > > **Response to Reviewer Htsg**
> > > > >
> > > > > Thank you for your recognition! If you would like to increase the score, we appreciate it, and this is just a gentle reminder that the score seems unchanged yet.

---

### Official Review · Reviewer_xuuN · 2022-07-11

**Rating:** 5
**Confidence:** 3
**Soundness:** 2 fair
**Presentation:** 3 good
**Contribution:** 2 fair

**Summary:**

The paper presents a framework CLEAR to generate counterfactual explanations on graphs. This framework can be helpful for promoting explainability in graph-based prediction models. Specifically, the authors use a graph variational autoencoder to generate the counterfactual graph and employ independent component analysis (ICA) to find causal relations. The generative counterfactual graphs have generalization ability and causality. Experiments show that CLEAR achieves promising performance in various evaluation metrics.

**Questions:**

1. The merits of ICA over the existing methods in identifying causality should be highlighted.
2. One important metric for explanations is human interpretability requiring the generated explanations should be compact. The authors should investigate human interpretability.
3. Counterfactual explanations can be used to defend against attackers. The authors are encouraged to conduct experiments on graphs generated by the adversarial attacks to verify the effectiveness of generated counterfactual explanations.
4. The experiments can not well support the claimed contribution in generalization. How to evaluate the generalization ability on unseen graphs? The work lacks the evaluation of generalization on unseen graphs.


**Limitations:**

There are no potential negative societal impacts.

**Strengths And Weaknesses:**

Originality: The paper employs two existing technologies, i.e., graph variational auto-encoder and ICA to generate the counterfactual graph. Although the two components are not new, the topic is novel. One of the contributions, optimization, is not original. Many works have employed various models to generate graphs by gradient-based optimization instead of enumeration, such as GNNExplainer, [1], and [2]. Existing works have proposed many methods, such as information flow [1] and Markov blanket [2], to identify causality among latent variables. What’s the merit of ICA over these existing methods?

Quality: Technically sound with well-supported claims.  However, the following issues are suggested to be further considered: 1. The evaluation metrics are not sufficient. One important metric for explanations is human interpretability requiring the generated explanations should be compact. The authors should investigate human interpretability. 2. In addition, counterfactual explanations may be used to defend against attackers. The authors are encouraged to conduct experiments on graphs generated by the adversarial attacks to verify the effectiveness of generated counterfactual explanations. 3. The work lacks the evaluation of generalization on the unseen graph.

Clarity: Overall, the paper is well written and easy to understand. However, the following issues need to be clarified. 1. The motivation behind employing ICA to find causality is not clear. 2. Why the team culture is not the cause of grant application in the example of the Preliminaries Section? 3. The grant application example for describing causality in counterfactual explanations is confusing. It is advisable to describe it in mathematical form.

Significance: The method proposed in this paper would be helpful to give counterfactual explanations and promote explainability in graph-based data.

[1] Lin, Wanyu, et al. "Orphicx: A causality-inspired latent variable model for interpreting graph neural networks." Proceedings of the IEEE/CVF Conference on Computer Vision and Pattern Recognition. 2022.

[2] Yang, Shuai, et al. "Learning causal representations for robust domain adaptation." IEEE Transactions on Knowledge and Data Engineering, 2021.

---

> ### Author Response · Authors · 2022-08-02
> **To Reviewer xuuN**
>
>
> Thank you for carefully reviewing our paper. We offer the following clarifications for your concerns:
>
> ### Q1: The merits of ICA over the existing methods in identifying causality should be highlighted.
>
> As far as we know, this is the first work which incorporates causality into counterfactual explanation on graphs. And the reason we take advantage of nonlinear ICA for better identifying causality is the natural connection between them [1]. The setting of nonlinear ICA and causality learning is essentially quite similar. Nonlinear ICA assumes that the observed data $X$ is generated by certain transformations on latent variables $Z$, and aims to identify $Z$ and the transformations. Similarly, in a causal model, the observed data is also generated based on a set of exogenous variables and structural equations. Considering this, we utilize the ICA techniques to promote causality in this paper.
>
> We’ve also carefully read the papers the reviewer mentioned (references [1,2] in the reviewer’s comments). Although these works also consider identifying causality, they are not targeting counterfactual explanation problems. These works focus more on identifying causal representations for model-based interpretation or domain adaptation, while in our problem we aim to generate counterfactual explanations which are compatible with the original structural causal model (SCM). The goals and scenarios of these problems are different.
>
> ### Q2: One important metric for explanations is human interpretability requiring the generated explanations should be compact.
>
> Thanks for this suggestion. Even though we did not explicitly evaluate the compactness of the explanation, in counterfactual explanation, we can show whether the explanations are compact through the proximity metric (i.e., the similarity between each original graph and its counterfactuals). Essentially, counterfactual explanation requires the counterfactuals to achieve a desired label with smallest perturbation on the original graph. In general, the higher the metric proximity is, the more compact, or sparse the perturbations should be.
>
> ### Q3: Counterfactual explanations can be used to defend against attackers. The authors are encouraged to conduct experiments on graphs generated by the adversarial attacks to verify the effectiveness of counterfactual explanations.
>
> Yes, counterfactual explanations can be used to defend against attackers [2]. Although it is not the main focus of this paper, It would be an interesting future work to further investigate their connection.
>
> ### Q4: The experiments can not well support the claimed contribution in generalization.
>
> We should clarify that the word “generalization” in this paper is different from “domain generalization” for graphs from different distributions. In this paper, the point is that our method can be directly applied for unseen graphs in an inductive way without retraining the model, while most existing graph CFE methods either are based on enumeration, or need to be trained separately for each input graph.
>
> In our experiments, we should clarify that the generalization ability has been implicitly validated. In Table 1, we compare the time cost of our method and baselines, as our method can be directly used for unseen graphs without retraining. Our method outperforms those baselines which need to be separately trained for unseen graphs (please refer to Line 297-302).

---

> > ### Author Response · Authors · 2022-08-02
> > **To Reviewer xuuN**
> >
> > ### Q5: Originality of optimization: Many works have employed various models to generate graphs by gradient-based optimization instead of enumeration.
> >
> > As far as we know, our work is the first gradient-based optimization method for counterfactual explanation on graphs without additional assumption of the prediction model and application domain.
> > Although gradient-based optimization has been employed in previous work of graph generation, in the area of counterfactual explanation on graphs, most existing methods are still in an enumeration way. GNNExplainer and the other two papers the reviewer mentioned are not counterfactual explanation methods (although some of them might be adapted for counterfactual explanation).
> >
> > For those few graph CFE methods which enable gradient-based optimization, most of them rely on specific domain knowledge [3] (e.g., chemical rules) or assumptions [4] about the prediction model (e.g., the prediction model is a graph neural network (GNN) model and its gradients or representations are accessible) to prune the search space or facilitate the optimization. However, these domain knowledge and assumptions limit their application in different scenarios, while our proposed method can optimize without these knowledge or assumptions.
> >
> >
> > ### Q6: The grant application example for describing causality in counterfactual explanations is confusing.
> >
> > In the real world, the causal model of grant application may be more complicated than the example we show. But the grant application process is just a hypothetical example to explain our motivation, so we choose a simple causal model for easier understanding.
> >
> > In this example, we do not assume the form of structural equations and data distributions in the causal model, so we did not fix it in mathematical form. In the example, the main point we try to convey is that, in order to achieve a desired prediction (e.g., grant approval), the counterfactual generator needs to perturb the original graph (e.g., increase the number of collaborations), but the perturbation needs to be consistent with the underlying causal relations between variables (e.g., more collaborations bring better team culture), otherwise, the counterfactual explanations might be unrealistic or even meaningless for human interpretation [5].
> >
> > [1] Khemakhem, Ilyes, et al. "Variational autoencoders and nonlinear ica: A unifying framework." International Conference on Artificial Intelligence and Statistics. PMLR, 2020.
> >
> > [2] Liu, Ninghao, et al. "Adversarial attacks and defenses: An interpretation perspective." ACM SIGKDD Explorations Newsletter 23.1 (2021): 86-99.
> >
> > [3] Numeroso, Danilo, and Davide Bacciu. "Meg: Generating molecular counterfactual explanations for deep graph networks." 2021 International Joint Conference on Neural Networks (IJCNN). IEEE, 2021.
> >
> > [4] Bajaj, Mohit, et al. "Robust counterfactual explanations on graph neural networks." Advances in Neural Information Processing Systems 34 (2021): 5644-5655.
> >
> > [5] Mahajan, Divyat, Chenhao Tan, and Amit Sharma. "Preserving causal constraints in counterfactual explanations for machine learning classifiers." arXiv preprint arXiv:1912.03277 (2019).

---

> > > ### Author Response · Authors · 2022-08-07
> > > **Looking forward to your further feedback**
> > >
> > > Dear Reviewer xuuN,
> > >
> > > Thank you again for your valued comments! We have responded to your initial questions, and we are looking forward to your further feedback. We will be happy to answer any further questions you may have.
> > >
> > > Thank you.
> > >
> > > Authors

---

> > > > ### Comment · Reviewer_xuuN · 2022-08-08
> > > > **Rebuttals Response**
> > > >
> > > > Thanks for the response. However, one of my concerns is not addressed. Why the similarity metric, i.e., proximity can measure compactness for interpretability? From equation (7), I do not think proximity can measure interpretability.

---

> > > > > ### Author Response · Authors · 2022-08-08
> > > > > **Response to Reviewer xuuN**
> > > > >
> > > > > Thanks for your additional feedback. For this concern, we clarify that: 1) counterfactual explanations promote model interpretation mainly through the comparison between the original graph and its counterfactuals, i.e., the perturbations which need to be made to achieve the desired prediction. 2) Proximity measures the similarity between the original graph and its counterfactuals. Intuitively, higher proximity indicates smaller perturbation, which leads the interpretation to be more compact.
> > > > >
> > > > > Please let us know if you have further questions or concerns, thanks!

---

### Meta-Review · Area_Chair_XKmr · 2022-08-23

**Recommendation:** Accept
**Confidence:** Certain

**Metareview:**

This paper proposes a new method for producing counterfactuals on graphs. This is performed using a VAE on graphs with auxiliary variables to identify independent components and promote causality. While this work is mainly a combination of existing ideas, the resulting method is not trivial.

The engaged discussion clarified most of the concerns except a remaining concern around the diversity of the explanations. The reviewer was encouraging to measure (or optimize for) the diversity of explanation. That is, explanations that are significantly different (e.g. orthogonal from each other) in latent space. This is not a ground for rejection but it could improve this work and we encourage the authors to add this feature.

I recommend acceptance of this paper.



**Award:**

No

---

### Decision · Program_Chairs · 2022-09-14

Accept